# ELEMENTARY: PATTERN-AWARE EVIDENCE DISCOVERY WITH LARGE LANGUAGE MODELS

## ABSTRACT

The remarkable success of rationale generation provokes precise Evidence Discovery, which aims to identify a small subset of the inputs sufficient to support a given claim. However, existing general extraction methods still fall short in quantifying the support of evidence and ensuring its completeness. This paper introduces a heuristic search framework, Elementary, which formulates the Evidence Discovery as a multi-step prompt construction process. Specifically, we offer a clear perspective that the LLMs prompted with *according to*, without fine-tuning on domain-specific knowledge, can serve as an excellent reward function to assess sufficiency. Based on this, Elementary explores various potential reasoning patterns and uses future expected rewards, including independent and pattern-aware rewards, to find the optimal prompt as evidence. Experiments on three common task datasets demonstrate that the proposed framework significantly outperforms previous approaches, additional analysis further validates that Elementary has advantages in extracting complex evidence.

## 1 INTRODUCTION

A key aspect of human intelligence lies in our capability to reason and solve complex problems (Negnevitsky, 2005). Recently, language models are steadily improving on making decisions and question-answering (Wang et al., 2019; Srivastava et al., 2022; Touvron et al., 2023; Team et al., 2024). But users still can't easily trust any given claim a model makes, because language models can hallucinate convincing nonsense (Maynez et al., 2020; Ji et al., 2023). To ensure trustworthiness and reliability, many rationalization methods focus on how to use evidence to yield prediction results, such as self-supported question-answering (Menick et al., 2022; Huang et al., 2024) and shortcuts discovery (Yue et al., 2024). Yet, high-quality evidence plays a critical role in trustworthy and explainable artificial intelligence, answering "*which part of the input should drive model to predict?*" (Evidence Discovery) is still a relatively unexplored task.

There are two tasks that are close to Evidence Discovery: Evidence Retrieval (Cartright et al., 2011; Bellot et al., 2013) and Evidence Detection (Rinott et al., 2015). However, Evidence Retrieval focuses on identifying *whole* documents, and Evidence Detection's goal is to pinpoint an independent text segment which can be used *directly* to support a claim, similar to Textual Entailment (Dagan et al., 2010). Additionally, although Evidence Discovery has been involved in fields such as summarization, fact-verification, and question-answering (Dou et al., 2021; Jiang et al., 2021; Zheng et al., 2024), there is still a lack of systematic research, most methods are task-specific, and require expensive manual annotation for supervised learning. The majority of existing approaches for Evidence Discovery adopt off-the-shelf embedding models or LLMs to retrieve relevant sentences from given input documents (Guo et al., 2022; Wang et al., 2024a; Zhu et al., 2023). Unfortunately, these methods have two obvious drawbacks. Firstly, relevant information may be insufficient to support the claim, existing methods ignore to evaluate sufficiency. Secondly, evidence typically doesn't appear in the form of a single sentence (Cattan et al., 2023). Previous work doesn't sufficiently capture the interactions between sentences when extracting evidence, limiting the exploration of potential reasoning patterns.

To address the evidence supportiveness problem, we turn to LLM reasoning with *according to* prompts (Weller et al., 2024). Recently, many works have demonstrated that LLMs can be effectively guided by natural language prompts (Ganguli et al., 2023; Wan et al., 2023). Inspired by

this, we attempt to use the *according to* prompt to ensure the model's grounding in context, in order to quantify the support of evidence for a given claim. Notably, we further verify that LLMs are sensitive to the strength of evidence support when guided by the *according to* prompt.

People explore different reasoning patterns by performing deductions in advance to discover chains of evidence that support a given claim. This process involves filtering, reorganizing, and integrating known information (Hattie & Jaeger, 1998). Inspired by this, we propose a pattern-aware heuristic search framework, named Elementary. Elementary formalizes evidence discovery as a multi-step prompting construction process, and uses LLMs with *according to* prompts to simultaneously evaluate independent and pattern-aware rewards. Based on this, Elementary can effectively explore more complete sets of evidence to support the given claims.

To validate the effectiveness of Elementary, we conduct experiments on three datasets, each from the areas of summarization, question-answering, and fact-checking, respectively. These scenarios challenge the generality of existing Evidence Discovery methods. Experimental results empirically show that Elementary consistently outperforms the competitive embedding-based and LLM-based baselines by a significant margin. Additionally, further analysis demonstrates that our method can capture deeper reasoning patterns, enabling more thorough Evidence Discovery.

## 2 RELATED WORKS

### 2.1 EVIDENCE DISCOVERY IN DIFFERENT TASKS

In many context-sensitive scenarios, developing a method to attribute claims is likely to be crucial for both system developers and users. For example, to obtain faithful abstractive summaries, previous studies (Dou et al., 2021; Wang et al., 2022; 2024b) attempt to find different types of guidance to support the output, Liu & Lapata (2019) uses a greedy algorithm to search for the evidence set most similar to the reference. In tasks such as generative question answering and fact-checking, many studies (Thorne et al., 2018; Augenstein et al., 2019; Su et al., 2021; Huang et al., 2023) commonly adopt a retrieval-enhanced framework: an evidence retriever is employed to query the background corpus for relevant sentences, to serve as evidence for the subsequent claim. However, even though evidence discovery has garnered widespread attention, most of methods are still *task-specific* and may require expensive *manual annotation* (Hanselowski et al., 2019; Kotonya & Toni, 2020; Zhang et al., 2023). In this paper, we argue this issue and propose a general Evidence Discovery framework to handle different scenarios.

### 2.2 EVIDENCE DISCOVERY BASED ON INFORMATION RETRIEVAL

Current approaches to identifying high-quality evidence typically adopt off-the-shelf retrieval models from the information retrieval (IR) field (Ma et al., 2019; Jiang et al., 2021; Chen et al., 2022). Existing retrieval methods can be broadly categorized into three types: statistical-based, embedding-based, and generative. Statistical-based methods, such as BM25 or ROUGE (Robertson et al., 2009; Liu & Lapata, 2019), rank a set of candidates based on the query terms appearing in each candidate, regardless of their proximity within the context. To address this issue, embedding-based methods use rich semantic features from pre-training. Embeddings make it possible to represent both candidates and claims as dense vectors in a high-dimensional semantic space and then use similarity score for nearest-neighbor retrieval (Soleimani et al., 2020; Wang et al., 2024a). However, this independent scoring paradigm fail to capture the interactions among sentences. Recently, generative models, particularly LLMs, have attracted an increasing amount of attention in the information retrieval field (Sun et al., 2023a; Qin et al., 2024). For example, Ma et al. (2023) and Sun et al. (2023b) design *listwise* prompt for document retrieval. Although prompted LLMs have improved retrieval accuracy by enabling more nuanced matching between queries and sources (Zhu et al., 2023), we remain skeptical about whether this sequence-to-sequence paradigm can effectively explore the organizational patterns within the evidence. Besides, it is also worth noting that the aforementioned retrieval method fails to consider the sufficiency and completeness of the evidence from a holistic perspective.

## 3 METHODS

### 3.1 TASK DESCRIPTION & FORMULATION

We introduce several concepts which will be used throughout this paper. **Claim**: a general, concise statement that something is the case, typically query-based or aspect-based. **Context**: a set of sentences potentially relevant to the claim, usually sourced from open-source news or articles. **Evidence**: any sentence of the context that supports or undermines the claim. For the purpose of this work, we assume that we are given a concrete claim $c$ and potentially relevant context $S = \{s_0, s_1, \ldots, s_n\}$, provided either manually or by automatic methods(Roush et al., 2024; Levy et al., 2014). The task, Evidence Discovery, aims to automatically extract an evidence set $E = \{e_0, e_1, \ldots, e_m\}$ from the unstructured context $\mathbb{S}$ that **support** the given claim $c$. It is worth noting that, unlike fact-checking(Thorne et al., 2018), Evidence Discovery assumes that the claim is partially or entirely correct based on the context.

We model the Evidence Discovery process as constructing multi-step prompts with optimal reasoning pattern, and introduce a heuristic search process to select evidence prompts step-by-step. Referring to the classical finite Markov Decision Process (MDP), we define the four ingredients of Elementary namely states, actions, transitions and rewards as follows: **State**: a state $o$ is a tuple $(c, \hat{E})$ for $c$ a claim and $\hat{E} = \{a_0, a_1, \ldots, a_k\}$ a set of sentences already selected from the context $S$. **Action**: an action $a$ is a sentence in the given context $S$. **Transition**: a transition $\mathcal{T}$ at step $t$ is a tuple $(o_t, a_t, o_{t+1})$, where $o_t = (c, \hat{E}_t)$, $o_{t+1} = (c, \hat{E}_{t+1})$ and $\hat{E}_{t+1} = \hat{E}_t \bigcup a_t$. **Reward**: the reward $\mathcal{R}$ for a transition $(o_t, a_t, o_{t+1})$ is to measure how well the claim $c$ is supported by $o_{t+1}$. Typically, we employ LLMs to generate policy $\pi(a_t|o_t) = P(a_t|o_t)$, where $a_t \in S - \hat{E}_t$. The policy $\pi$ tends to select candidates related to the preceding context, which helps maintain consistency in reasoning. In practice, we also introduce a length penalty to balance candidates of different lengths. Based on the LLM policy $\pi$, the value of transition $(o_t, a_t, o_{t+1})$ is given by a Q-function:

$$Q_\pi(o_t, a_t) = \mathbb{E}_\pi \left[ \sum_{k=0}^{K} \gamma^k \mathcal{R}(a_{t+k}, o_{t+k}) \right]. \tag{1}$$

Then, following the Bellman equation, the optimal policy $\pi^\star$ of the MDP process should satisfy:

$$Q_{\pi^\star}(o_t, a_t) = \mathcal{R}(a_t, o_t) + \gamma \max_{a_{t+1} \in S - \hat{E}_{t+1}} Q_{\pi^\star}(o_{t+1}, a_{t+1}). \tag{2}$$

### 3.2 QUANTIFY THE SUPPORT OF EVIDENCE USING *according to* PROMPT

Before introducing the Elementary formally, we discuss how to quantify the support of an input for a target claim, which is the foundation of Elementary. When making decisions, or engaging in critical analysis, humans typically organize and integrate information to logically derive specific conclusions, a process known as deductive reasoning. Similarly, the answer generation process of common LLMs is autoregressive, where the prediction of the next token depends on the previous context. Therefore, this work assumes that LLMs are excellent deducers, capable of accurately perceiving the sufficiency of evidence prompt: the more logical the prompt, the greater the likelihood that the LLM will generate the target claim.

> **Model Input**
>
> **Claim:** Girlfriends was published more frequently than the magazine founded by Edward L. Youmans.
> **Context:** (1) Popular Science is an American bi-monthly magazine carrying popular ... (2) Popular Science has won over 58 awards, including the American Society of ...
>
> **Grounding via *according-to* Prompting:**
>
> Context: **<Context>** , according to the given context, we can infer that **<Claim>** .

Figure 1: Prompting LLMs to ground in context.

However, considering that LLMs may tend to produce outputs that deviate from the input, known as hallucination or inconsistency, we first introduce *according-to* prompts to ground LLMs' output in a given context $\hat{S}$. Figure 1 shows the proposed prompt. Then, we force LLMs to decode the given claim $c$ and directly compute the log probability as score, where $score(c, \hat{S}) = \sum_1^{|c|} \log P(c_i | c_{<i}, \text{prompt}(\hat{S}))$.

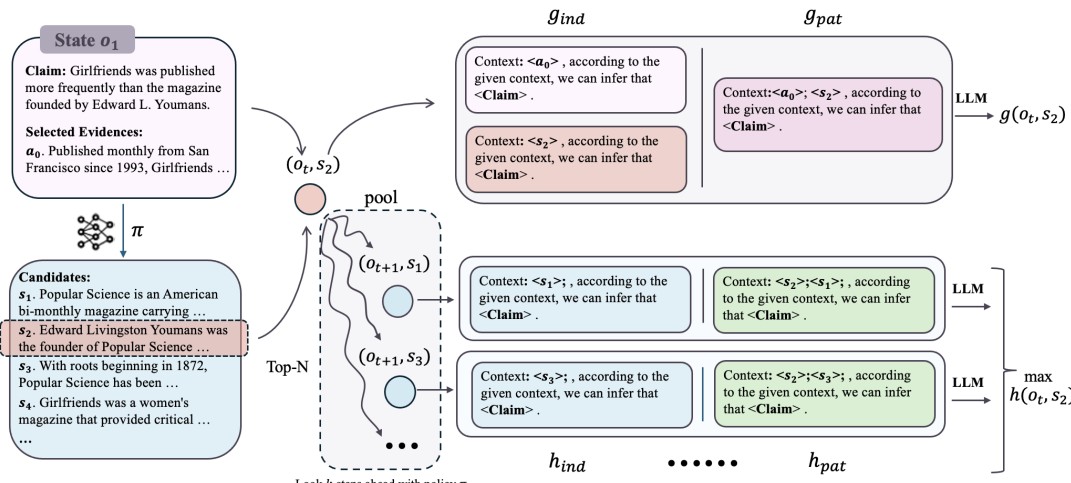

Figure 2: An illustration of value function $f$. Here, we set k=1 for ease of demonstration.

## 3.3 ELEMENTARY: PATTERN-AWARE HEURISTIC SEARCH

Elementary uses a value function $f$ to approximate the real $Q$-function, aming to overcome the vast and complex search space. Unlike previous approaches that rely on supervised learning to fit the Q-function, based on section 3.2, we design an unsupervised value function to evaluate the reward of taking action $a_t$ in the state $o_t$. Specifically, $f$ is defined as:

$$f(o_t, a_t) = g(o_t, a_t) + \gamma h(o_t, a_t), \tag{3}$$

where $g(o_t, a_t)$ represents the cumulative reward of state $o_t$ after taking action $a_t$, and $h(\mathbf{a}_t)$ denotes a heuristic function for estimating the expected future reward of taking action $a_t$. Besides, $\gamma$ is a discount factor used to balance the importance of $g(\cdot)$ and $h(\cdot)$.

**Cumulative Reward.** As shown in equation 4, the cumulative reward $g(o_t, a_t)$ consists of two parts: $g_{ind}(o_t, a_t)$, assessing the independent contribution of each $a_{t'}$ to $c$ in a context-independent manner; $g_{pat}(o_t, a_t)$, concatenating $a_t$ with $a_{0:t-1}$ to explore the "chemical reaction" between $a_t$ and the selected evidence, evaluate the current reasoning patterns. We use $\lambda$ to balance $g_{ind}(\cdot)$ and $g_{pat}(\cdot)$.

$$g(o_t, a_t) = g_{ind}(o_t, a_t) + \lambda g_{pat}(o_t, a_t)$$
$$s.t. \quad \begin{cases} g_{ind}(o_t, a_t) = \frac{1}{t} \sum_{t'=0}^{t} score(c, a_{t'}) \\ g_{pat}(o_t, a_t) = score(c, a_{0:t}) \end{cases} \tag{4}$$

**Future Reward.** A heuristic function $h(o_t, a_t)$, similar to $g(o_t, a_t)$, is introduced to estimate the potential future benefit of taking action $a_t$. As shown in Figure 2, starting from the state-action pair $(o_t, a_t)$, we perform rollout with policy $pi$ to form a trajectory pool, representing different reasoning patterns. In practice, we usually select the top-N trajectories to approximate the solution. Then, the highest future reward of the best reasoning pattern is regarded as the potential value of taking action $a_t$. The purpose of $h(o_t, a_t)$ is to provide guidance on which unselected context sentences might, together with $(o_t, a_t)$, form a reasoning pattern that strongly supports the given claim $c$. In equation 5, $K$ is a hyperparameter used to determine how many steps to look ahead, and $\delta$ is a balancing factor. By using this function, our search framework can prioritize exploring states that appear to be closer to the end goal, thus reducing the overall search time and making the search process more efficient.

---

**Algorithm 1** Framework of pattern-aware Evidence Discovery.

---

**Input:**
    Claim $c$; the set of context sentences, $S$;
    LLM policy $\pi$; the maximum evidence size, $max\_step$.
**Output:**
    Evidence $\hat{E}$.
1: Initialize $\hat{E}_0 \leftarrow \emptyset$; $o_0 \leftarrow (c, \hat{E}_0)$; $t \leftarrow 0$.
2: **while** $t \leq max\_step$ **do**
3:    $f_{values} \leftarrow dict()$
4:    **for** $s_i$ in $\pi(\cdot|\hat{E}_t, S)$ **do**
5:       $g(o_t, s_i) \leftarrow g_{ind}(o_t, s_i) + \lambda g_{pat}(o_t, s_i)$
6:       $\hat{a}_{t+1:t+K} \leftarrow \underset{\mathcal{T} \sim \pi(\cdot|\hat{E}_t \bigcup s_i, S)}{\arg\max} \sum_{k=1}^{K} \gamma^{k-1}(h_{ind}(o_{t+k}, \mathcal{T}_k) + \delta h_{pat}(o_{t+k}, \mathcal{T}_k))$
7:       $h(o_t, s_i) \leftarrow \sum_{k=1}^{K} \gamma^{k-1}(h_{ind}(o_{t+k}, \hat{a}_{t+k}) + \delta h_{pat}(o_{t+k}, \hat{a}_{t+k}))$
8:       $f_{values}[s_i] \leftarrow g(o_t, s_i) + \gamma h(o_t, s_i)$
9:    **end for**
10:   update $a_t \leftarrow \underset{s_i}{\arg\max} f_{values}[s_i]$; $\hat{E}_{t+1} \leftarrow \hat{E}_t \bigcup a_t$; $o_{t+1} \leftarrow (c, \hat{E}_{t+1})$; $t \leftarrow t + 1$
11: **end while**
12: **return** $\hat{E}_t$;

---

$$h(o_t, a_t) = \max_{\substack{\mathcal{T} \sim \pi \\ a_{t+k} \in \mathcal{T}}} \sum_{k=1}^{K} \gamma^{k-1}(h_{ind}(o_{t+k}, a_{t+k}) + \delta h_{pat}(o_{t+k}, a_{t+k}))$$

$$s.t. \quad \begin{cases} h_{ind}(o_t, a_t) = score(c, a_{t+k}) \\ h_{pat}(o_t, a_t) = score(c, a_{t:t+k}) \end{cases} \tag{5}$$

Algorithm 1 give a overview of Elementary. Specifically, Elementary uses a greedy strategy to determine how to expand the current evidence prompts. At each iteration of the main loop, we associate each candidate $s_i$ with a $f$-value estimating how much reward will be attained if we expand $s_i$, and the candidate with the highest $f$-value is selected to update state $o_t$. The algorithm continues until a specified number of sentences are selected.

# 4 EXPERIMENTS

## 4.1 SETTING

### 4.1.1 DATASETS

Ideal test dataset should meet three conditions: first, we hope the claims are completely or partially correct, facilitating the search for supporting sentences; second, the claims should have a certain level of abstraction, requiring contextual reasoning with a reasoning path length greater than 1; finally, the test datasets should cover multiple domains to test the generalizability of the methods. Based on this, we conduct experiments on three common benchmarks, including HoVer (Jiang et al., 2020), PubMedQA (Jin et al., 2019), and CovidET (Zhan et al., 2022). Among them, HoVer is a multi-hop dataset with manually annotated evidence, ensuring the claims are abstract. However, since HoVer was originally designed for fact-checking, the claims may not be correct. Therefore, we randomly selected 200 instances labeled as true for testing. Besides, PubMedQA is a generative question-answer dataset in the biomedical field, while CovidET is an abstract summarization dataset in the COVID-19 domain. Both tasks require a deep understanding of the context to generate answers; therefore, we consider the reference answers as claims. However, since both datasets lack evidence annotations, we selected 200 instances from each dataset for manual annotation.

Table 1: Results on HoVer, PubMedQA and CovidET Datasets.

| Method | HoVer | | | PubMedQA | | | CovidET | | |
|---|---|---|---|---|---|---|---|---|---|
| | P | R | F1 | P | R | F1 | P | R | F1 |
| **Top-3** | | | | | | | | | |
| ROUGE | 57.3 | 52.4 | 54.8 | 39.0 | 45.1 | 41.7 | 44.3 | 41.3 | 42.8 |
| BM25 | 58.0 | 53.1 | 55.4 | 40.0 | 46.7 | 43.1 | 48.7 | 45.3 | 46.9 |
| MPNet-base | 58.7 | 53.7 | 56.1 | 44.7 | 52.1 | 48.1 | 53.7 | 50.0 | 51.8 |
| GTE-large | 60.0 | 54.9 | 57.3 | 46.0 | 53.7 | 49.6 | 55.7 | 51.9 | 53.7 |
| Gemma-Retriever | 59.7 | 54.6 | 57.0 | 41.6 | 48.3 | 44.7 | 55.3 | 51.5 | 53.4 |
| Gemma-Reranker | 61.0 | 55.8 | 58.3 | 45.3 | 52.8 | 48.8 | 56.7 | 52.8 | 54.7 |
| RankGPT | 62.2 | 55.2 | 58.5 | 43.3 | 50.6 | 46.7 | 55.9 | 51.9 | 53.8 |
| Elementary | 64.7 | 59.2 | 61.8 | 48.0 | 55.6 | 51.4 | 60.7 | 56.6 | 58.5 |
| **Top-5** | | | | | | | | | |
| ROUGE | 47.0 | 69.8 | 56.2 | 34.0 | 66.5 | 45.2 | 37.6 | 58.4 | 45.7 |
| BM25 | 48.9 | 72.6 | 58.4 | 33.4 | 65.0 | 44.1 | 38.4 | 59.6 | 46.7 |
| MPNet-base | 49.5 | 73.5 | 59.1 | 36.0 | 68.8 | 47.3 | 43.8 | 68.0 | 53.3 |
| GTE-large | 50.5 | 75.0 | 60.3 | 36.7 | 70.1 | 48.1 | 43.6 | 67.7 | 53.0 |
| Gemma-Retriever | 52.2 | 72.9 | 60.8 | 34.9 | 66.2 | 45.7 | 43.1 | 63.5 | 51.4 |
| Gemma-Reranker | 51.0 | 75.7 | 60.9 | 37.1 | 70.8 | 48.7 | 43.7 | 67.9 | 53.2 |
| RankGPT | 53.6 | 79.6 | 64.0 | 35.9 | 68.1 | 47.0 | 44.1 | 65.2 | 52.6 |
| Elementary | 55.2 | 82.0 | 66.0 | 38.0 | 73.5 | 49.9 | 45.4 | 70.5 | 55.2 |

### 4.1.2 IMPLEMENTATION DETAILS

We use Gemma-2b-it[1] to generate the policy $\pi$ and quantify support, its advantages lie in its lightweight design and strong inference performance. The implementation of our framework based on transformers library[2]. Specifically, the hyperparameters $\gamma$, $\delta$, and $\lambda$ are set to 0.9, 1, and 1, respectively. When exploring potential reasoning patterns to obtain the maximum future reward, we look ahead $K = 4$ steps and calculate the $N = 10$ paths with the highest probabilities. All experiments were conducted on a 6xRTX3090 machine with 16-bit quantization enabled. All decoding/sampling settings were kept default. Following previous works, we use Precision, Recall and F1 score as the evaluation metrics for Evidence Discovery (Zhang et al., 2023).

### 4.1.3 BASELINES

we select several representative general extraction methods as baselines:

- ROUGE (Chin-Yew, 2004): count the number of overlapping units between the candidates and the given claim.

- BM25 (Robertson et al., 2009): rank candidates based on the claim term occurrence and rarity across the whole context.

- MPNet (Song et al., 2020): use the all-mpnet-v2-base version[3] to calculate the similarity between the sentence embeddings of each candidate and the given claim.

- GTE (Li et al., 2023): a general text embedding model trained with multi-stage contrastive learning, we use GTE-large[4] to calculate the candidate-claim similarity.

- Gemma-Retriever: concatenate all candidate sentences as input and prompts Gemma-7b-it[5] to directly generate the top-k most relevant sentences.

---

[1]https://huggingface.co/google/gemma-2b-it

[2]https://github.com/huggingface/transformers

[3]https://huggingface.co/sentence-transformers/all-mpnet-base-v2

[4]https://huggingface.co/thenlper/gte-large

[5]https://huggingface.co/google/gemma-7b-it

Table 2: Quantifying the strength of evidence support.

|                  | -w/o *according to* | -w *according to* |
|------------------|--------------------|-------------------|
| not related      | -4.3625            | -4.4688           |
| not relevant     | -4.2188            | -4.1875           |
| sufficient       | -3.2344            | -2.9464           |
| -w/o 1 sentence  | -3.5000            | -3.2656           |
| -w/o 2 sentences | -3.8594            | -3.7188           |
| -w/o 3 sentences | -4.0312            | -3.9844           |
| -w/o 4 sentences | -4.3125            | -4.4062           |
| -w/ not related  | -3.2656            | -2.9862           |
| -w/ not relevant | -3.1106            | -2.9672           |

- Gemma-Reranker: concatenate all sentences that pass the initial filter by the GTE-large model as input and prompts Gemma-7b-it to rerank these candidates.
- RankGPT (Sun et al., 2023b): similar to gemma-retriever, a listwise prompting-based approach using GPT-3.5-turbo.

## 4.2 MAIN RESULTS

We start by evaluating the effectiveness of Elementary on three general benchmarks. Table 1 compares its performance with state-of-the-art baselines under the topk-3 and topk-5 settings. We highlight three key observations: 1). Elementary consistently outperforms various evaluated baselines across different tasks. In contrast, none of the baseline approaches consistently perform well across all three datasets. 2). The statistical-based methods perform the worst when the claims are relatively abstract. The LLM-based methods, such as Gemma-Retriever and RankGPT, do not significantly outperform the embedding-based methods. On the PubMedQA dataset, the performance of LLM-based methods is even markedly lower than that of embedding-based methods. 3). The Elementary framework executed with Gemma-2b-it significantly outperforms the Gemma-Retriever and Gemma-Reranker based on Gemma-7b-it, achieving up to 3.8%-6.7% higher F1 score than Gemma-Retriever and 3.8%-6.7% higher F1 score 1.2%-5.1% than Gemma-Reranker.

## 5 ANALYSIS

### 5.1 ARE LLMS SENSITIVE TO THE DEGREE OF SUPPORT FOR EVIDENCE?

Previous works have demonstrated that LLMs can be prompted to calculate the relevance between two sentences (Qin et al., 2024). However, these scoring methods often lack a point of reference, making it difficult to quantify the variations in the degree of support. In this section, we verify that the output probability given by the LLM with *according to* prompt can serve as an effective metric for quantifying evidence support. As shown in Table 2, We categorize the input into the following cases based on the degree of support it provides for the claim: 1) not related. Randomly select $m$ sentences from contexts unrelated to the given claim as input; 2) not relevant. Randomly select $m$ non-evidence sentences from the context corresponding to the given claim; 3) sufficient. Concatenate all sentences in the golden evidence set as input; 4) -w/o $m$ sentences. Randomly remove $m$ sentences from the set of golden evidence, and concatenate the remaining sentences as input. as input; 5) -w/ not related. Add sentences from the not related set to the set of golden evidence; 6) -w/ not relevant. Add sentences from the not relevant set to the set of golden evidence. We report the average log probability (token-level) of each claim.

Based on the results shown in Table 2, we have the following findings: 1) Without introducing additional input noise, the LLM can accurately perceive the sufficiency of the evidence, regardless of whether the *according to* prompt is used. However, after using the *according to* prompt, this perception becomes more sensitive and shows greater fluctuations; 2) The *according to* prompt helps LLMs to perceive related but irrelevant noise; 3) The feedback from the LLM prompted with *according to* aligns with human performance on different inputs, making it an ideal reward function.

Table 3: Performance on 1/2/3/4-hop data.

| Method | FEVER-1 | | HoVer-2 | | HoVer-3 | | HoVer-4 | |
|---|---|---|---|---|---|---|---|---|
| | F1 | EM | F1 | EM | F1 | EM | F1 | EM |
| ROUGE | 45.0 | 45.0 | 63.0 | 41.0 | 55.7 | 16.5 | 59.5 | 10.0 |
| BM25 | 51.0 | 51.0 | 68.5 | 47.5 | 59.3 | 17.0 | 59.0 | 10.0 |
| MPNet-base | 52.5 | 52.5 | 69.0 | 46.5 | 61.3 | 16.0 | 59.3 | 10.5 |
| GTE-large | 50.0 | 50.0 | 73.0 | 53.0 | 59.0 | 16.5 | 59.8 | 13.0 |
| Gemma-Retriever | 59.5 | 59.5 | 65.0 | 43.0 | 55.3 | 14.5 | 56.0 | 9.5 |
| Gemma-Reranker | 62.0 | 62.0 | 71.0 | 50.5 | 63.0 | 19.0 | 53.0 | 11.0 |
| RankGPT | 70.0 | 70.0 | 68.5 | 46.5 | 61.7 | 18.5 | 63.0 | 14.0 |
| Elementary | 61.0 | 61.0 | 76.0 | 57.5 | 66.3 | 26.0 | 68.8 | 21.5 |

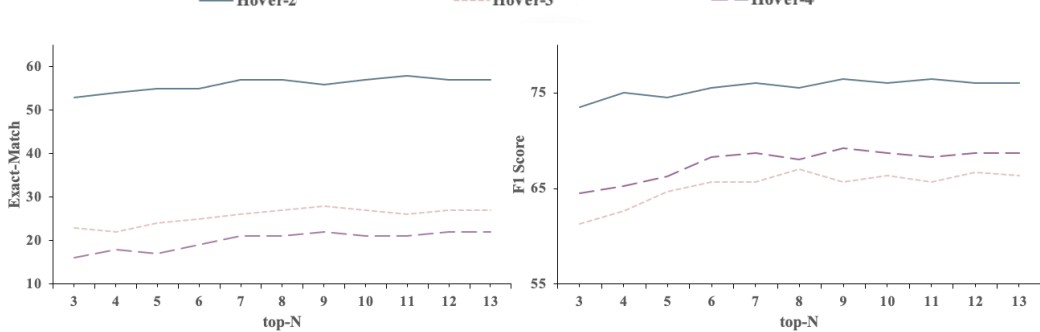

Figure 3: Performance comparison on the HoVer dataset under different size of the trajectory pools.

## 5.2 IS ELEMENTARY A GENERAL-PURPOSE EVIDENCE DISCOVERY METHOD?

Table 1 demonstrates that Elementary exhibits a clear advantage over the mainstream embedding-based and LLM-based extraction methods across different tasks and domains. Here, we further validate that Elementary can extract evidence of varying complexity. Specifically, we categorize the HoVer test set based on the number of evidence corresponding to each claim, and then randomly select 200 examples from each category for testing. We also conduct experiment on the 1-hop FEVER dataset (Thorne et al., 2018). In addition to the F1 score, we also report the Exact-Match (EM) score to assess the ability of each method to extract complete evidence. Our method shows significant improvement in extracting complex evidence, with greater improvement as the number of hops increases. Additionally, in the 1-hop scenario, Elementary can achieve satisfactory performance using only the independent reward.

## 5.3 HOW DOES THE SIZE OF THE TRAJECTORY POOL AFFECT PERFORMANCE?

Elementary uses a rollout policy $\pi$ for expansion. A larger trajectory pool represents more candidate paths but increases inference cost. In Figure 3, we compare the performance of our Elementary across different sizes of the trajectory pools, using the 2/3/4-hop HoVer datasets. We highlight two key observations: 1). At the initial stage, the performance of evidence extraction improves as the number of candidate reasoning paths increases. 2). The more complex the evidence, the slower its corresponding curve converges.

## 5.4 HOW DOES THE CHOICE OF BASE MODEL AFFECT PERFORMANCE?

In this section, we discuss the impact of model size and instruction fine-tuning on the performance of the proposed framework. The experimental results on the HoVer dataset are shown in Figure 4. Specifically, we compared the performance of Gemma-2b, Gemma-2b-it, Gemma-7b, and Gemma-

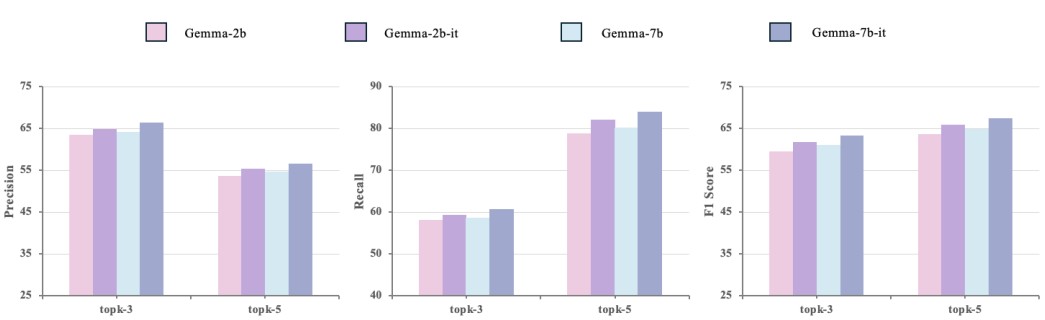

Figure 4: Performance comparison on the HoVer dataset under different number of rollouts.

Table 4: Performance of ablation study.

|  | FEVER-1 | | HoVer-2 | | Hover-3 | | Hover-4 | |
|---|---|---|---|---|---|---|---|---|
|  | F1 | EM | F1 | EM | F1 | EM | F1 | EM |
| Elementary | 61.0 | 61.0 | 76.0 | 57.5 | 66.3 | 26.0 | 68.8 | 21.5 |
| -w/o *according to* | 54.0 | 54.0 | 72.0 | 53.0 | 59.0 | 20.0 | 65.3 | 18.5 |
| -w/o *pattern* | 61.0 | 61.0 | 73.5 | 51.5 | 60.3 | 19.5 | 63.0 | 17.5 |
| -w/o *independent* | 61.0 | 61.0 | 75.5 | 55.5 | 63.7 | 24.5 | 67.0 | 22.0 |
| -w/o $h(\cdot)$ | 61.0 | 61.0 | 74.5 | 52.0 | 62.0 | 21.0 | 65.3 | 18.0 |
| -w/o $\pi$ | 61.0 | 61.0 | 76.0 | 54.0 | 65.7 | 24.0 | 68.3 | 20.5 |

7b-it under the top-3 and top-5 settings. We found that instruction fine-tuning yields a more significant performance improvement than merely increasing the model size. This is likely because instruction fine-tuning enhances the model's ability to follow prompts effectively.

## 5.5 ABLATION ANALYSIS

We design ablation studies to verify the effectiveness of core modules. As shown in Table 4, removing the *according to* prompt results in the worst performance, indicating that it plays a key role in Elementary. Comparatively, removing the independent rewards ($g_{ind}$ and $h_{ind}$) achieves superior performance on EM metric over removing the pattern-aware rewards ($g_{pat}$ and $h_{pat}$), demonstrating that the pattern-aware rewards are particularly advantageous for sufficient Evidence Discovery. Besides, the future reward $h(\cdot)$ is also important for extracting complex evidence. Finally, planning reasoning paths with policy $\pi$ performs better than random selection.

## 6 CONCLUSION

In this paper, we highlight the importance of the task of Evidence Discovery and its distinction from similar tasks. We argue that current general extraction methods struggle to accurately quantify the strength of evidence and ensure its completeness. Therefore, we present a heuristic search framework called Elementary, which treats Evidence Discovery as a multi-step prompt construction process. Specifically, we verify that LLMs, when prompted with *according to*, can act as an effective reward function to evaluate sufficiency. Based on this, we introduce pattern-aware future reward to explore potential optimal reasoning paths. Experiments across three common task datasets show that our framework significantly surpasses previous methods, and further analysis confirms Elementary's strength in extracting complex evidence completely. We also realize that our framework has certain limitations. For example, its input length is constrained by the maximum positional encoding of LLMs, which hinders fine-grained evidence discovery in long text environments, we will explore this question in the future. Nevertheless, we believe that Elementary can enhance awareness of evidence discovery and facilitate rationale generation in various domains.

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
