# OpenReview forum: "Elementary: Pattern-aware Evidence Discovery with Large Language Models"
_ICLR.cc/2025/Conference — ICLR 2025 Conference Withdrawn Submission_

### Official Review · Reviewer_c34m · 2024-10-30

**Soundness:** 3
**Presentation:** 2
**Contribution:** 2
**Rating:** 5
**Confidence:** 4

**Summary:**

The authors introduce a new framework for Evindent Discovery overcoming the previous baseline regarding the completeness of the evidence. They take advantage of heuristic search which provides independent and pattern-aware rewards. Their pipeline is based on a multi-step prompt construction process that includes "according to" to ensure grounding the LLM in the context and sufficiency.

**Strengths:**

- The use of a heuristic reward function to filter different pathways regarding selecting evidence seems interesting and similar to a look-ahead strategy to prune irrelevant pathways.

**Weaknesses:**

- I am not sure how your works compare to chain-of-thought/evidence and what advantages it brings w.r.t CoT. It would be interesting if you could include CoT for comparison purposes or at least clarify if it is relevant or not. For this reason, I think your contribution is limited.
- Section 5.4 doesn't have a significant contribution as you only indicated instruction fine-tune models are better in instruction following which is obvious.
- According to Table 3 and Fig. 3, the drop in the EM score from 2-hop to 3-hop is significant which points out regarding the pattern recognition it is not very powerful when the number of supporting evidences is more than 2. Therefore, although it is better than baselines, but it still seems weak to say it is pattern-aware.

* The EM score for the FEVER-1 in Table 3 is a duplication of the F1 score. The same thing happens in Table 4.
* Please highlight the best scores in your tables to make it easy for comparison.
* TYPO: in line 117, you need to put a space after "methods".

**Questions:**

I have no question.

---

### Official Review · Reviewer_sfjs · 2024-11-01

**Soundness:** 2
**Presentation:** 2
**Contribution:** 2
**Rating:** 3
**Confidence:** 3

**Summary:**

The paper presents a formulation and an heuristic search framework to rank evidence from context provided to an LLM for generation using the "according to" prompting mechanism.

**Strengths:**

Investing on better mechanism to rank evidence, passages and context for LLMs to answer prompts is an area of research that is relevant and will eventually provide better benchmarks and better use of LLMs.

Section 5 and its analysis posed as questions is good an informative.

**Weaknesses:**

- The paper is difficult to follow and does not set the context right for the reader. For example, in the abstract, it is unclear what "according to" in cursive means without reading the paper. This happens in lines 52 (where there is at least a citation) and 61 again.

- it is very unclear why the baselines in Table 1 are the right ones? Please motivate. See my comments about citing relevant literature in RAG, maybe that would help motivating whether these are the right baselines in the future version of the paper.

- The authors claim that elementary consistently outperforms various evaluated baselines, but it is unclear if the results are statistically significant and whether the improvements are relevant to the reranking of passages/context. Also, the impact of with or without according to is unclear based on table 2, which repots the log probability of each claim is not fully clear to me.

- I suggest to provide more information in the captions of the tables in the paper. They break the flow of reading the paper, and how they connect with the tasks and datasets. Note that the calculation of metrics is never explained. Similarly, why top-3 and top-5 are the right settings for evaluation is never explained.

-The choice of LLM (Gemma-2b-it) is not well motivated. Why this one, why not another one? The paper later discusses this in Section 5.4, but the writing is unclear as it is not preparing the reader for that in line 294

See my questions below as support for above comments.

minor: - when you say "typically, we employ LLMs..." cite a paper that shows that this is typical.

**Questions:**

I'm surprised to see that in the task description and formulation the context and the evidence is not directly connected with retrieval augmented generation which has become a very active area of research in recent years;  the authors chose to to only cite 2014 papers for this (note that I am not saying they should not be cited, just complement with recent work where this can be related). You can start by citing Retrieval-Augmented Generation for Knowledge-Intensive NLP Tasks - Part of Advances in Neural Information Processing Systems 33 (NeurIPS 2020) / Retrieval-Augmented Generation for Large Language Models: A Survey
Yunfan Gao, Yun Xiong, Xinyu Gao, Kangxiang Jia, Jinliu Pan, Yuxi Bi, Yi Dai, Jiawei Sun, Meng Wang, Haofen Wang - Why not connecting with this?

The task of evidence discovery where an evidence set is extracted from the context sounds quite similar to the retrieval step in RAG using an embedding model that ranks passages by similarity that are then provided as context to an LLM for generation. I understand that the methodology and claims are different, but I wonder if that would be a good way to connect with the readers familiar with RAG, and even experiment with datasets recently used here. Why not connecting with this?

Similarly, quantifying the support of an input for a target claim sounds very similar to the idea of citations and source attribution in RAG. Why not connecting with this?

In line 160, I don't understand how you are "forcing" the LLM to decode the given claim. Providing an "according-to" prompt will not alleviate hallucinations, it will just alleviate them. You even show that in Section 5.5. This, again, shows that the paper needs to improve in its writing. I suggest to not use the word "forcing", maybe "guiding"

have you tried variations of the wording for "according to the given context, we can infer that <Claim>"? there are multiple ways to write this, and this may impact the final outcome.

---

### Official Review · Reviewer_GByr · 2024-11-03

**Soundness:** 3
**Presentation:** 2
**Contribution:** 3
**Rating:** 6
**Confidence:** 3

**Summary:**

The paper presents Elementary, a heuristic search framework for evidence discovery that leverages large language models through multi-step prompts to identify supportive evidence for given claims. This work aims to tackle the problem that existing work lacks in identifying the supportiveness of the retrieved evidence. Therefore, they introduce a framework that includes “pattern-aware” and “independent” reward mechanisms to enhance evidence sufficiency and completeness without domain-specific fine-tuning. Their framework was evaluated on two gemma models with HoVer, PubMedQA, and CovidET datasets. Their results show that Elementary outperforms baseline methods by capturing complex evidence patterns across diverse domains.

**Strengths:**

Their Elementary framework formulates evidence discovery as a multi-step prompt construction process using LLMs. This innovative approach combines heuristic search methods with LLM-guided pattern exploration, which is unconventional compared to standard single-step or retrieval-based techniques.
Introducing the “according to” prompt and forcing the LLM to decode the given claim and calculate the log probability as the score for assessing evidence sufficiency is interesting. They designed an unsupervised value function considering both sentence and paragraph-level evidence dependence rewards.

**Weaknesses:**

Base model generalizability: The author did their experiments on gemma models with two different parameter sizes. The value function’s result on the same test instance could be similar since both models are trained on similar lexical. Have the authors considered evaluating their framework with other models of similar size to show the generalizability across different base models?
Computational cost: This framework requires multiple computations between the evidence and the claim checked. Have the author compared the computational cost between the baseline method and their framework?

**Questions:**

1. The notation for **a** and **s** is a bit confusing to me. Does the **s** represent the potentially relevant context in the paragraph/document, and **a** stands for the single sentence from **s**?
2. Since the author mentioned, "All decoding/sampling settings were kept default."-- Are there any variances in the predicted results in different runs?
Suggestion: Please consider including more details in the figure and table captions. Figure 3's caption is very clear and easy to understand.

---

### Note · Authors · 2024-11-26

I have read and agree with the venue's withdrawal policy on behalf of myself and my co-authors.